# Native lattice strain induced structural earthquake in sodium layered oxide cathodes

Gui-Liang Xu [1✉], Xiang Liu[1], Xinwei Zhou[2], Chen Zhao[1], Inhui Hwang[3], Amine Daali[1,4], Zhenzhen Yang[1], Yang Ren[3,6], Cheng-Jun Sun[3], Zonghai Chen [1], Yuzi Liu [2✉] & Khalil Amine [1,5✉]

High-voltage operation is essential for the energy and power densities of battery cathode materials, but its stabilization remains a universal challenge. To date, the degradation origin has been mostly attributed to cycling-initiated structural deformation while the effect of native crystallographic defects induced during the sophisticated synthesis process has been significantly overlooked. Here, using in situ synchrotron X-ray probes and advanced transmission electron microscopy to probe the solid-state synthesis and charge/discharge process of sodium layered oxide cathodes, we reveal that quenching-induced native lattice strain plays an overwhelming role in the catastrophic capacity degradation of sodium layered cathodes, which runs counter to conventional perception—phase transition and cathode interfacial reactions. We observe that the spontaneous relaxation of native lattice strain is responsible for the structural earthquake (e.g., dislocation, stacking faults and fragmentation) of sodium layered cathodes during cycling, which is unexpectedly not regulated by the voltage window but is strongly coupled with charge/discharge temperature and rate. Our findings resolve the controversial understanding on the degradation origin of cathode materials and highlight the importance of eliminating intrinsic crystallographic defects to guarantee superior cycling stability at high voltages.

[1] Chemical Sciences and Engineering Division, Argonne National Laboratory, Lemont, IL 60439, USA. [2] Centre for Nanoscale Materials, Argonne National Laboratory, Lemont, IL 60439, USA. [3] X-ray Science Division, Argonne National Laboratory, Lemont, IL 60439, USA. [4] University of Wisconsin-Milwaukee, 3200 North Cramer Street, Milwaukee, WI 53211, USA. [5] Materials Science and Engineering, Stanford University, Stanford, CA 94305, USA. [6] Present address: Department of Physics, University of Hong Kong, Kowloon, Hong Kong. ✉email: xug@anl.gov; yuziliu@anl.gov; amine@anl.gov

The emerging demand for high-energy and low-cost batteries for electric vehicles and grid-scale energy storage application calls for rapid improvements in cathode materials[1]. Lithium/sodium-layered transition metal (TM) oxides have attracted tremendous attention as appealing cathode materials because of their high specific capacities[2,3]. To further increase the energy density, a prevailing approach is to push the charging voltage limit to simultaneously attain higher specific capacity and increase the average working voltage[4]. However, these layered cathodes undergo universal capacity drop and voltage decay during high-voltage operation[5–7]. Over several decades, extensive fundamental understanding and material development have been carried out to reveal the underlying failure mechanism and mitigate the structural degradation at elevated voltage.

Irreversible surface/bulk phase transition upon cycling has been reported as one of the prevalent origins for the performance degradation of layered cathodes[8,9]. Several prominent studies showed that surface reconstruction, such as layered to spinel/rock-salt, can initiate from the cathode surface, and then gradually propagate into the bulk structure during a high-voltage charge. This eventually causes bulk fatigue of lithium layered oxide cathodes after long-term cycling[10,11]. Compared with their Li analogs that present mostly the octahedral structure through direct synthesis, sodium-layered oxide cathodes can be classified into P-type (prismatic) and O-type (octahedral), depending on the surrounding Na environment and the number of unique oxide layers[12,13]. Therefore, in general, they exhibit much more complex phase transitions such as P2-O2[14], P2-Z[15], P2-OP4[16], and O3-P"3[17] during high-voltage charge, leading to irreversible bulk structural changes and huge volume changes. In both cases, researchers agree that the irreversible phase transition can lead to accumulation of mechanical stress at the phase boundaries due to lattice mismatch and further intergranular/intragranular cracking of the cathode particles after prolonged cycling[18–20]. Therefore, tremendous efforts have been focused on suppressing the undesired phase transition through aliovalent doping[21–25].

The parasitic reactions between charged cathodes and electrolytes have also been correlated with the high-voltage instability of layered oxide cathodes[26]. On the one hand, the common electrolytes solvents such as ethylene carbonate, diethyl carbonate, and dimethyl carbonate are thermodynamically unstable at high voltage due to their limited electrochemical stability window[27]. On the other hand, the charged cathodes containing highly oxidized transition metals will aggravate the decomposition of electrolytes and lead to the formation of thick cathode–electrolyte-interphase (CEI) on the cathode surface and hence sluggish $Li^+/Na^+$ diffusion[28]. Mu et al. have reported that the cathode–electrolyte interfacial reaction can also trigger transition metal reduction/dissolution, heterogeneous surface reconstruction, and nanocracks[29]. To address these concerns, surface coating[30–33] and high-voltage electrolytes[34–36] have been widely developed to enhance the high-voltage cycling stability of layered cathodes.

These valuable findings are acknowledged, but the aforementioned degradation mechanism has been mainly attributed to the dynamic structural changes (e.g., phase transition, interfacial reactions, and mechanical cracks) that nucleated and evolved during cycling. In fact, the explicit role of these structural degradations and an accurate assignment on the degradation origin remain challenging because they are intimately coupling with each other during electrochemical charge/discharge. In contrast, the effect of native structural defects often induced during the complicated synthesis process of layered oxides has been significantly overlooked. Many trial-and-error efforts have shown a strong correlation between synthetic conditions and

battery performance of layered oxides, and yet the fundamental mechanism remains elusive. Moreover, cathode materials with tailor-made defects through precise synthetic control could serve as a model structure to probe their explicit role in battery performance.

Here, by using in situ synchrotron X-ray diffraction (SXRD) to probe the solid-state synthesis process of sodium-layered oxide cathodes, we intentionally synthesized a highly strained O3 $NaNi_{0.4}Mn_{0.4}Co_{0.2}O_2$ cathode. The electrochemical voltage window control experiments in combination with in situ SXRD, synchrotron X-ray absorption spectroscopy, and differential electrochemical mass spectrometry during charge/discharge unanimously revealed that the irreversible phase transition and cathode–electrolyte reactions are not the dominant factor for the rapid capacity fade of strained $NaNi_{0.4}Mn_{0.4}Co_{0.2}O_2$ cathode during cycling. Instead, using advanced transmission electron microscopy (TEM), we discovered that the native high lattice strain plays an overwhelming role in triggering the destructive structural earthquake of sodium-layered cathodes, which spontaneously relaxed due to local strain heterogeneity and led to severe breakdown/fragmentation of layered structure during prolonged cycling. We also confirmed that its relaxation process is strongly coupling with the operating temperature and charge/discharge rate. We believe this work is the first to decouple the contribution of native lattice strain to high-voltage instability of sodium-layered cathodes and counters the conventional wisdom that phase transition and cathode–electrolyte reactions are primarily responsible for degradation.

## Results

**In situ synthesis study of strained $NaNi_{0.4}Mn_{0.4}Co_{0.2}O_2$ by SXRD.** Generally, the synthesis of layered oxides involves solid-state reactions between transition metal hydroxides precursors and lithium/sodium salts by heating their mixture at a high temperature for a sufficiently long time, followed by a cooling process. Recently, researchers leveraged in situ SXRD and computational modeling to investigate the evolution of non-equilibrium kinetic intermediates and the formation of thermodynamic equilibrium phases during these processes[37,38]. Although these studies provide valuable guidance for the predictive synthesis of layered oxides, most of them have focused on examining the impact of structural characteristics during the heating/holding process, such as phase impurity, lattice parameter, crystallite size, Li–TM bond length, and $Li^+/Ni^{2+}$ mixing[39–41]. The effect of microstrain evolution during the cooling process, especially under rapid quenching, has been largely ignored. Notably, our previous study showed that appropriate quenching could help to stabilize the metastable P2/O1/O3 intergrowth phase and tailor the electrochemical performance[42]. Among various phase structures in the sodium-layered oxides cathodes, the O3 phase represents the structure with higher Na content, which is different from the P2-type cathode that has a lower Na content and suffers from sodium deficiency problem[13,43]. Here, we further explored the formation process of O3 $NaNi_{0.4}Mn_{0.4}Co_{0.2}O_2$ cathode and revealed the effect of quenching-induced lattice strain on the crystal structures and electrochemical performance.

Figure 1a and Supplementary Fig. 1a show the 2D contour plot of in situ SXRD patterns during the formation process of O3 $NaNi_{0.4}Mn_{0.4}Co_{0.2}O_2$. The starting material was a mixture of $Ni_{0.4}Mn_{0.4}Co_{0.2}(OH)_2$ and 5% excess mole of NaOH. As shown more clearly by the covariance analysis and the corresponding thermalgravimetric analysis (TGA) in Supplementary Fig. 1b, c, the major reactions started to occur at around 150 and 350 °C, respectively. Therefore, the formation process of O3

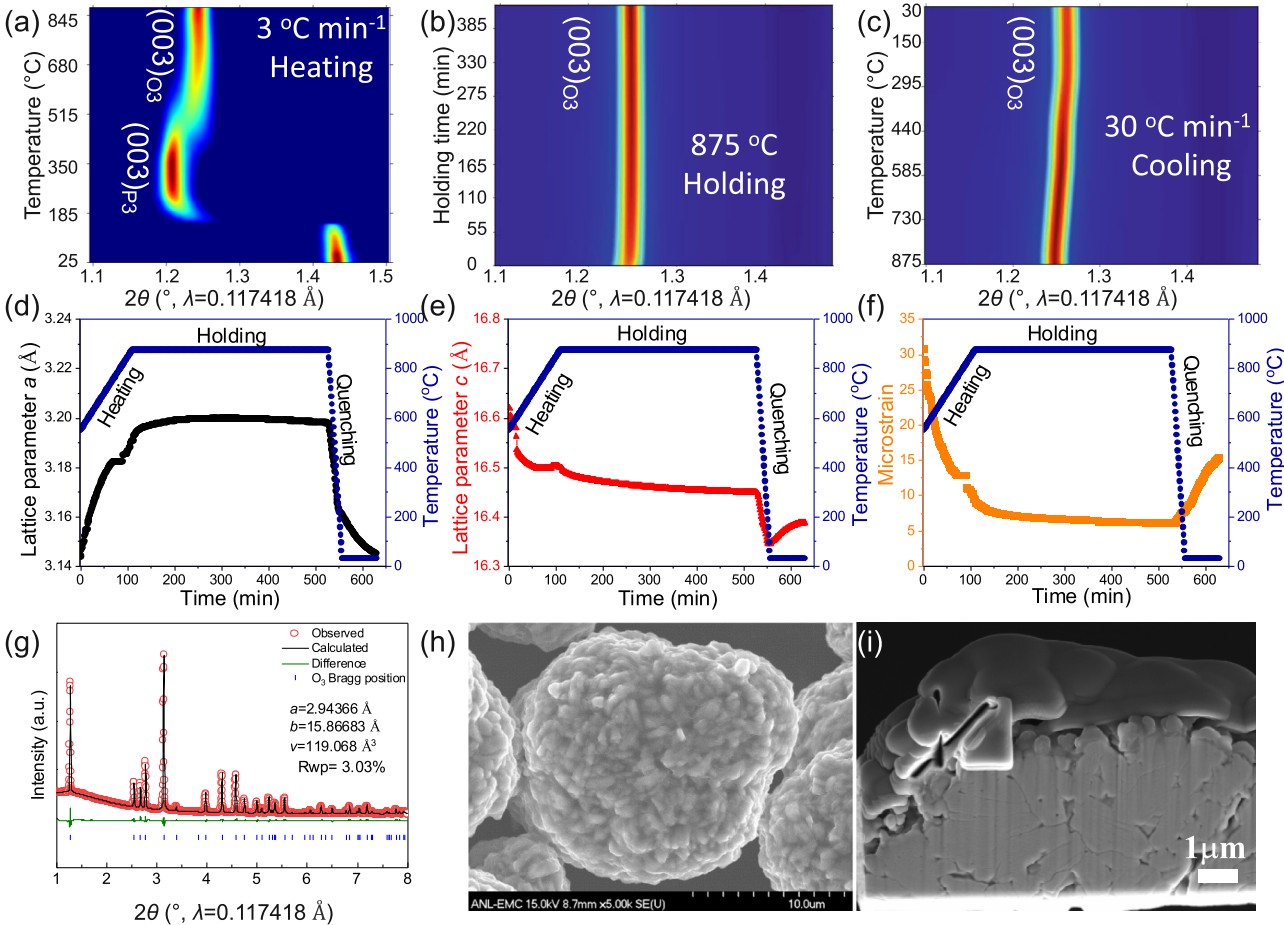

**Fig. 1 Solid-state synthesis of O3 NaNi$_{0.4}$Mn$_{0.4}$Co$_{0.2}$O$_2$.** 2D contour plot of in situ SXRD patterns during **a** heating, **b** holding, and **c** quenching process. The color in **a–c** represents the intensity, with red for highest and blue for lowest. **d** Lattice parameter $a$, **e** lattice parameter $c$ and **f** microstrain evolution during the heating/holding/quenching process. **g** SXRD Rietveld refinement, **h** SEM image and **i** cross-section SEM image of the strained O3 NaNi$_{0.4}$Mn$_{0.4}$Co$_{0.2}$O$_2$.

NaNi$_{0.4}$Mn$_{0.4}$Co$_{0.2}$O$_2$ can be classified into three regions according to their phase composition: 25–185 °C (region I, starting material), 185–500 °C (region II, intermediates), and 500–875 °C (region III, final product). The Rietveld refinement in Supplementary Fig. 2 shows that the starting material can be indexed well as Ni(OH)$_2$ and NaOH, while the XRD pattern of the intermediate product at 350 °C is consistent with that of P3-Na$_{0.8}$Ni$_{0.4}$Mn$_{0.4}$Co$_{0.2}$O$_2$. Upon further reaction beyond 500 °C, the structure of the O3 phase started to evolve and dominate. The O3 phase remained during the holding and quenching process (Fig. 1b, c).

In order to understand subtle structural changes during the reaction process, Rietveld refinement was performed on all the measured SXRD patterns after the evolution of the O3 phase (i.e., 500 °C). As shown in Fig. 1d, e, the lattice parameter $a$ was dramatically increased during heating and then gradually stabilized during the holding process; in contrast, the lattice parameter $c$ was significantly decreased during heating and then slowly decreased during the holding process. These results were attributed to the gradual transformation of P3 ($a = b = 2.8479$ Å, $c = 16.6093$ Å) to O3 ($a = b = 2.94$ Å, $c = 15.923$ Å) phase during heating, which exhibited distinct cell lattice parameter. In general, the structural difference between different phases can induce non-uniform lattice strain (microstrain) in the crystallite. Therefore, as shown in Fig. 1f, the decrease of P3/O3 phase boundaries during the heating/holding process led to a gradual reduction of the microstrain.

We found that quenching is necessary to maintain the O3 phase structure during cooling. With a slow cooling under an air atmosphere, the obtained material exhibited a mixed phase of P1/P2/O3 (Supplementary Fig. 3), which might be due to the surface reconstruction induced by slow cooling[39]. The CO$_2$ molecules in the air would coordinate with Na$^+$ at the solid/gas interface or attack the O atom of NaO$_6$ octahedra, leading to the extraction of surficial Na$^+$ from sodium-layered oxide cathodes and hence the formation of Na$_2$CO$_3$ and Na-deficient phase, such as P1 and P2[39]. Indeed, a clear signal of Na$_2$CO$_3$ can be observed in the C 1$s$ X-ray photoelectron spectroscopy (XPS) spectrum of a slow-cooled sample (Supplementary Fig. 4). In sharp contrast, the quenched sample exhibited high purity O3 phase, demonstrating a good fit with the standard O3 layered structure (Fig. 1g). The morphologies of the quenched cathode are shown in Fig. 1h, i, and exhibited characteristic conventional polycrystalline features.

Surprisingly, as shown in Fig. 1d, e, the lattice parameter $a$ and $c$ were both decreased due to the lattice shrinkage when abruptly exposed to a low temperature. During the relaxation process of the inhomogeneous thermal stress in the quenched sample, lattice parameter $a$ was significantly decreased while $c$ was drastically increased, which is similar to what occurs during the de-sodiation process of layered oxides. This indicates an increase of structural defects such as vacancies, dislocations, and stacking faults. Quenching-induced strain generation has been well known in steel manufacturing[44]. It has been also often used for the fabrication of battery cathode materials to control grain size,

oxygen vacancies, and etc[45,46]. During quenching, a highly inhomogeneous temperature field is generated, which can result in heterogeneous thermal stresses, thus leading to residual stresses being introduced at the end of the quenching process. In our experiment, the quenching was conducted under air, which can easily induce temperature heterogeneity in the local lattice region, leading to the formation of the observed lattice strain. Therefore, the microstrain during the rapid quenching process exhibited an intense increase (Fig. 1f); while the effect of such quenching-induced native lattice strain on the electrochemical performance of layered cathodes has not been well understood.

**Electrochemical characterization of strained $NaNi_{0.4}Mn_{0.4}Co_{0.2}O_2$.** The electrochemical performance of the as-prepared cathode was evaluated by using half cells with sodium metal as referenced and countered electrode. The electrolyte was 1 mol/L $NaPF_6$ in propylene carbonate with a 2 vol% fluoroethylene carbonate additive. Figure 2a shows the first charge/discharge curves

of the strained $O3-NaNi_{0.4}Mn_{0.4}Co_{0.2}O_2$ cathode within 2.0–4.4 V at 0.08C (1C = 180 mA/g), which exhibited the characteristic features of the O3 phase with multiple-step voltage plateaus. The first charge and discharge capacity were measured to be 180.6 and 171.9 mAh/g, respectively, leading to a high initial Coulombic efficiency of 95%. However, upon further charge/discharge, the capacity exhibited a continuous decrease. After 100 cycles, the reversible capacity was only 54 mAh/g, resulting in a low capacity retention of 30% (Fig. 1b).

Such degradation has been previously attributed to irreversible structural transformation or parasitic reactions at high voltage[4–6,26,29]. Figure 2c displays the in situ differential electrochemical mass spectrometry (DEMS) result of the strained $O3-NaNi_{0.4}Mn_{0.4}Co_{0.2}O_2$ cathode during charge/discharge within 2.0–4.4 V. Although $O_2$ gas release has been often observed in the layered oxide cathodes when charged to high-voltage, it is also possible for the transformation of the oxygen species to $CO_2$ because of the chemical reactions between reactive lattice oxygen and electrolytes at high-voltage[47–49]. Indeed, despite no evolution

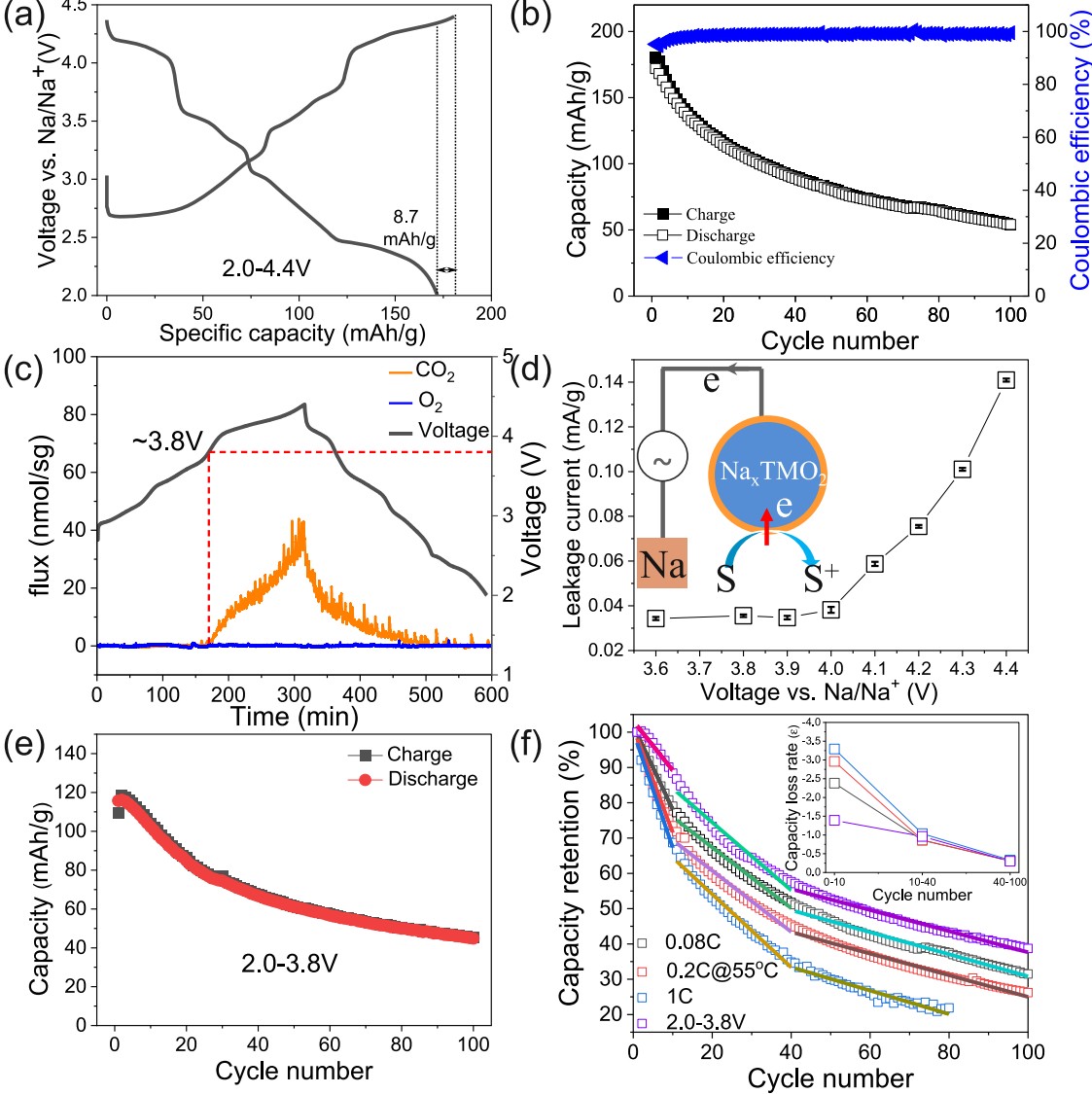

**Fig. 2 Electrochemical characterization. a** 1st charge/discharge curve and **b** cycling performance of strained O3 $NaNi_{0.4}Mn_{0.4}Co_{0.2}O_2$ cathode within 2.0–4.4 V at 0.08C. **c** In situ DEMS and **d** parasitic reaction study of O3 $NaNi_{0.4}Mn_{0.4}Co_{0.2}O_2$ cathode. S and S⁺ in **d** represent solvents and oxidized solvents, respectively. **e** Cycling performance of strained O3 $NaNi_{0.4}Mn_{0.4}Co_{0.2}O_2$ cathode within 2.0–3.8 V at 0.08C. **f** Normalized capacity retention of O3 $NaNi_{0.4}Mn_{0.4}Co_{0.2}O_2$ cathode at different cycling conditions. The capacity loss rate in **f** is the slope of the linear fitting curve of the capacity retention curve.

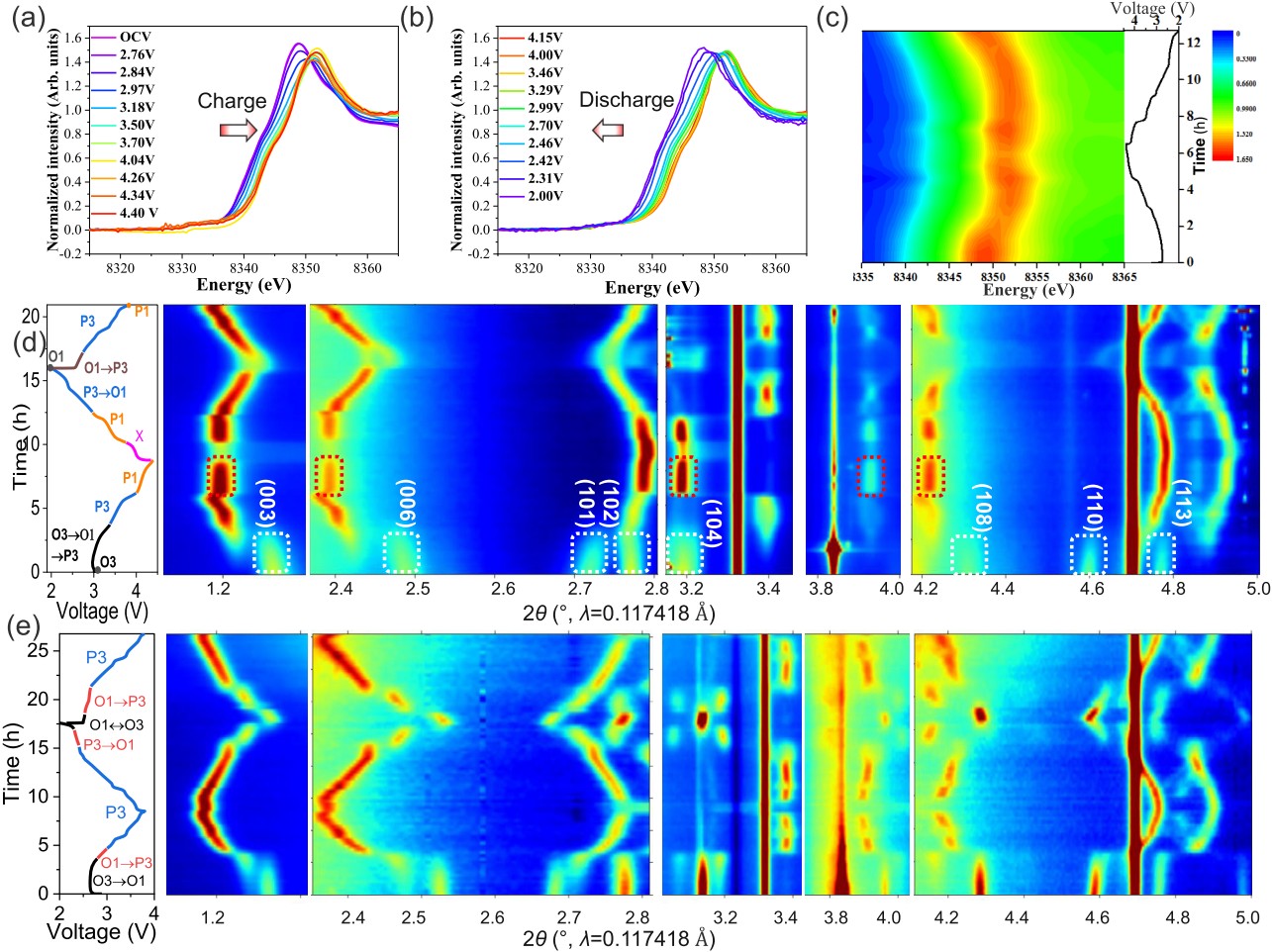

**Fig. 3 In situ synchrotron characterization during charge/discharge.** In situ Ni K-edge XANES of strained O3 NaNi$_{0.4}$Mn$_{0.4}$Co$_{0.2}$O$_2$ within 2.0–4.4 V: **a** during charge, **b** during discharge, and **c** 2D contour plot during charge/discharge. Voltage profiles and the corresponding 2D contour plot of in situ SXRD patterns during charge/discharge of O3 NaNi$_{0.4}$Mn$_{0.4}$Co$_{0.2}$O$_2$ cathode within **d** 2.0–4.4 V and **e** 2.0–3.8 V. The color in **c**–**e** represent the intensity, with red for highest and blue for lowest.

of O$_2$ gas during the whole charge process, upon charging beyond 3.8 V, a significant release of CO$_2$ can be clearly observed. This should come from either decomposition of electrolytes or side reactions between oxygen radicals and electrolytes[49,50]. Consistent with this observation, the high-precision leakage current measurement results (Fig. 2d) showed that the parasitic current increased dramatically when the cut-off voltage was increased to over 4.0 V. Therefore, in order to mitigate the capacity degradation induced by high-voltage charge, the strained O3-NaNi$_{0.4}$Mn$_{0.4}$Co$_{0.2}$O$_2$ cathode was cycled between 2.0 and 3.8 V. Surprisingly, the cycle stability was not improved by lowering the charge cut-off voltage, which contradicts results in other studies of reported sodium-layered oxide cathodes with narrow voltage window[51]. A capacity retention of only 38.7% was attained after 100 cycles (Fig. 2e). Moreover, the charge/discharge test of the highly strained NaNi$_{0.4}$Mn$_{0.4}$Co$_{0.2}$O$_2$ cathode at an elevated voltage or temperature demonstrated similar rapid degradation (Supplementary Fig. 5). Interestingly, the capacity fade rate (inset in Fig. 2f), which were obtained by linear fitting of the capacity retention curve, is almost the same after the first 10 cycles, regardless of voltage window, testing temperature and charge/discharge rate, whereas they exhibited significantly different results for the first 10 cycles. The results indicate that the capacity degradation of the highly strained NaNi$_{0.4}$Mn$_{0.4}$Co$_{0.2}$O$_2$ cathode might be controlled by a specific factor, while such factor was

regulated by the operating condition in the early stage of charge/discharge.

**Redox couple evolution and phase transition of strained NaNi$_{0.4}$Mn$_{0.4}$Co$_{0.2}$O$_2$ during charge/discharge.** In situ Ni K-edge X-ray absorption near-edge spectroscopy (XANES) characterization was carried out to understand the redox couple evolution of the highly strained NaNi$_{0.4}$Mn$_{0.4}$Co$_{0.2}$O$_2$ cathode during charge/discharge within 2.0–4.4 V, which can provide the oxidation state changes of Ni element during battery operation. As shown in Fig. 3a, b, the Ni K-edge shifted to high-energy upon extraction of Na$^+$ due to the oxidation of Ni$^{2+}$ to Ni$^{3+}$/Ni$^{4+}$ during charge, while it shifted back to lower energy during discharge due to the reduction of Ni$^{3+}$/Ni$^{4+}$ to Ni$^{2+}$[42]. In general, as shown in the 2D contour plot of Ni K-edge XANES throughout the whole charge/discharge process (Fig. 3c), the Ni undergoes a highly reversible Ni oxidation/reduction during charge/discharge within 2.0–4.4 V. Even after cycling for 10 cycles that have triggered severe capacity degradation in the strained O3 cathode, Ni still exhibits reversible redox reactions (Supplementary Fig. 6a), which cannot explain the severe capacity loss of strained NaNi$_{0.4}$Mn$_{0.4}$Co$_{0.2}$O$_2$ cathode reported here (Fig. 2b). The redox reaction behavior of Co and Mn in the O3 strained NaNi$_{0.4}$Mn$_{0.4}$Co$_{0.2}$O$_2$ cathode were also explored and both

showed reversible transformation during cycling within 2.0–4.4 V (Supplementary Fig. 6b, c), indicating that the observed capacity degradation is not from the cycling-induced structural transformation.

Therefore, we further performed in situ SXRD to reveal the phase transition of strained $NaNi_{0.4}Mn_{0.4}Co_{0.2}O_2$ cathode with a controlled voltage window. Figure 3d shows the 2D contour plot of in situ SXRD patterns during charge/discharge within 2.0–4.4 V, in accompaniment with the corresponding voltage curve and phase transition process. As shown, the electrode before charge/discharge can be well indexed using the O3 phase. Upon charge, the (003) and (006) peaks shifted toward lower angles, which indicated an expansion of the $c$ lattice parameter due to the increased oxygen electrostatic repulsion between oxygen layers induced by the removal of $Na^+$. Meanwhile, the (101), (102), (110), and (113) peaks moved towards higher angles during charge, corresponding to shrinkage of $a$ lattice parameter due to the oxidation of TM. In addition, the intensity of the O3 phase gradually decreased, while the peaks of the O1 phase started to appear. Upon further charge, the O1 phase was then transformed into the P3 phase starting at 3.37 V. At the long plateau beyond 4.0 V, the structure of the P1 phase started to dominate the charged product. At the end of the charge process, an unknown X phase with low Na content and crystallinity was formed. Therefore, the phase transformation of strained $NaNi_{0.4}Mn_{0.4}Co_{0.2}O_2$ cathode during high-voltage charge can be described as O3 → O1 → P3 → P1 → X, which is similar to the previously reported O3 sodium-layered cathode when charged to high voltage[52]. During the discharge process, the phase transformation process reversed. However, at the beginning of the discharge process, the XRD intensities of (00$l$) peaks are very weak and broad, indicating severe lattice strain at the $c$ axis direction that prevents the re-insertion of $Na^+$. Moreover, as evidenced by the disappearance of O3 (003), (006), (101), (108) and (110) peaks of the O3 phase, the O1 phase (rather than the original O3 phase) dominated the fully discharged electrode, indicating an irreversible phase transition during high-voltage cycling.

In sharp contrast with that charge/discharge within 2.0–4.4 V, the highly strained $NaNi_{0.4}Mn_{0.4}Co_{0.2}O_2$ cathode exhibited a highly reversible phase transformation of O3 ↔ O1 ↔ P3 with a lower charge cut-off voltage of 3.8 V. As clearly shown in Fig. 3e, the in situ SXRD patterns during the whole charge/discharge process presented a highly symmetric feature, and all the peaks of O3 phase were fully recovered at the end of the discharge process. Such a reversible phase transition during charge/discharge of layered oxide cathodes is often considered favorable for the stabilization of their cycle performance[53]. However, we illustrate that this is not the case of the highly strained $NaNi_{0.4}Mn_{0.4}Co_{0.2}O_2$ cathode (Fig. 2e). Thus, its degradation origin remains elusive.

**Native lattice strain-induced structural earthquake in $NaNi_{0.4}Mn_{0.4}Co_{0.2}O_2$.** Inspired by the in situ SXRD results during the quenching process, we further examined the crystallographic structure of the highly strained $NaNi_{0.4}Mn_{0.4}Co_{0.2}O_2$ cathode at the atomistic level by using advanced TEM. Figure 4a, b shows the low and high magnification TEM image of the pristine $NaNi_{0.4}Mn_{0.4}Co_{0.2}O_2$ particle, respectively. Unlike the slow-cooling synthesized O3 sodium oxide cathodes that showed smooth surface and well-aligned lattice fringes[54], the quenched cathode exhibited a highly rough surface and fluctuated strain contours in a large area. Interestingly, we do not observe an obvious composition heterogeneity across the strained region (Supplementary Fig. 7). These structural features have been mostly observed in the cycled cathodes materials[1,55], but barely in the pristine cathodes. The HRTEM image in Fig. 4c illustrates that the $d$-spacing along the $c$ axis direction is about 0.549 nm, which is stretched by about 3.58% compared to that of the standard O3 phase (0.530 nm). A closer examination (inset of Fig. 4c) clearly shows the existence of severe lattice distortion along the $c$ axis direction, in which we can see the curved lattice fringe and overlap of TM atoms and Na atoms. Such lattice distortion might accelerate cation mixing or cation migration during cycling, leading to undesired structural evolution and

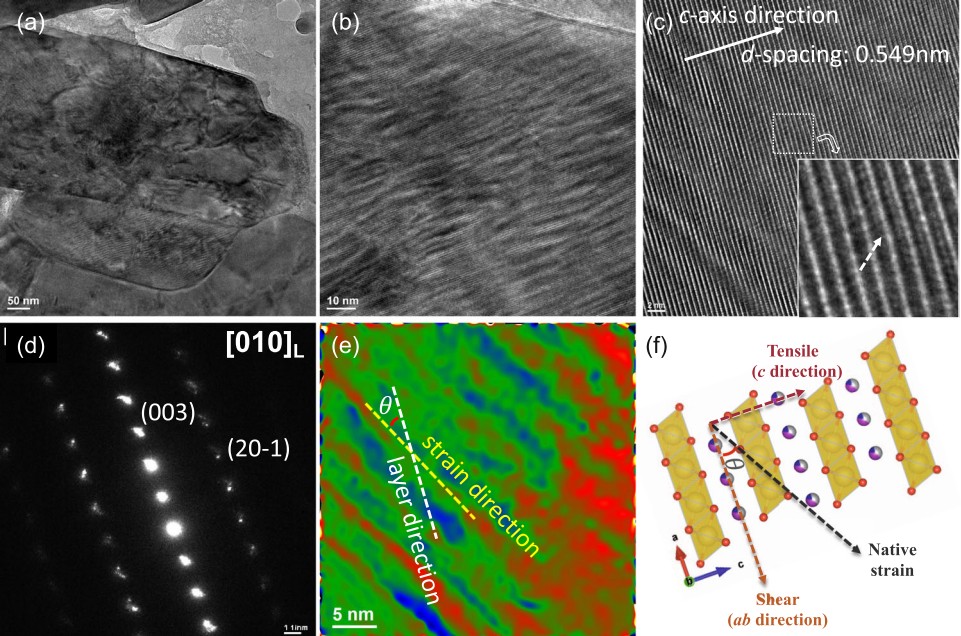

**Fig. 4 TEM characterization of pristine O3 $NaNi_{0.4}Mn_{0.4}Co_{0.2}O_2$. a** Low and **b** high magnification, **c** high-resolution bright-field TEM image, **d** SAED pattern, **e** GPA analysis and **f** atomic structural model of the strained O3 $NaNi_{0.4}Mn_{0.4}Co_{0.2}O_2$. Inset in **c** is the zoomed-in view of the region marked by a white square. The color in **c** represents the intensity, with red for highest and blue for lowest. $\theta$ is the angle between layered direction and strain direction. The yellow, red, blue, gray, and purple spheres in **f** represent Na, O, Ni, Co, and Mn atoms, respectively.

hence capacity/voltage fade[56–59]. In addition, the $d$-spacing along the $ab$ axis exhibited an inhomogeneous distribution (Supplementary Fig. 8). The selected area electron diffraction (SAED) pattern in Fig. 4d is in good agreement with the [010] projection of layered O3 cathode, but exhibits obvious spot splitting. All the aforementioned structural features are due to the native high lattice strain induced during the quenching process, leading to the highly metastable nature of strained O3 cathode. The corresponding geometric phase analysis (GPA) of Fig. 4c provided the direct strain distribution of the $NaNi_{0.4}Mn_{0.4}Co_{0.2}O_2$ cathode, which showed an angle between layered direction and strain direction (Fig. 4e). Such native lattice strain can be thus separated into tensile stress along the $c$ axis direction that tends to stretch the lattice, and shear stress along the $ab$ axis direction that will compress the lattice (Fig. 4f). Upon extraction of Na⁺, such metastable structure tends to undergo spontaneous strain relaxation and cause significant structural degradation as revealed below.

Figure 5a shows the bright-field TEM image of strained $NaNi_{0.4}Mn_{0.4}Co_{0.2}O_2$ cathode after charge/discharge within 2.0–4.4 V at 0.08C for 100 cycles. Compared to the pristine one (Fig. 4b), a large portion of the native lattice strain was relaxed since the strain contours disappeared. HRTEM image (Fig. 5b) and the zoomed-in view (Fig. 5c) shows that there are numerous stacking faults and dislocations as a catastrophic consequence of strain relaxation. In particular, we can clearly see the bending of lattice (yellow dashed lines in Fig. 5c) parallel to the layered

direction due to the shear stress along the $ab$ axis. Moreover, the tensile stress along the $c$ axis direction led to the evolution of lattice dislocations and stacking faults. As a result, it will be difficult to re-insert Na⁺ into the Na layer because of the crossover of TM cations that might occupy the Na sites and damage layered structure, implied by the disappearance of (00$l$) peaks in the in situ SXRD patterns at the beginning of the discharge process (Fig. 3d). Hence, the material suffered from a dramatic capacity loss.

On the other hand, by examining the strain-relaxed region in Fig. 5a (marked by a white rectangle), it was found that the observed gaps between two layered planes are not empty; they are composed by low-crystalline fragmented domains (Supplementary Fig. 9). This conclusion is further supported by the STEM-EDS elemental mapping of cycled cathode in Supplementary Figs. 10 and 11, in which the electrolytes (signal element of P) cannot penetrate into the outlined gaps while the signal of TM, such as Ni, can be clearly observed. Figure 5d clearly showed that a layer of NiO-like rock-salt structure with (111) plane was formed at the tip of the premature crack region. As reported by Wang and co-workers, the (111) plane of rock-salt structure is energetically and structurally favorable, and has a high tolerance against compression strain[60]. However, the native high lattice strain in the quenched cathode significantly exceeds both tensile and compression limit of the rock-salt phase, thus leading to the formation of rock-salt fragments with different orientations (Fig. 5e and supplementary Fig. 9). In contrast, in the strain-

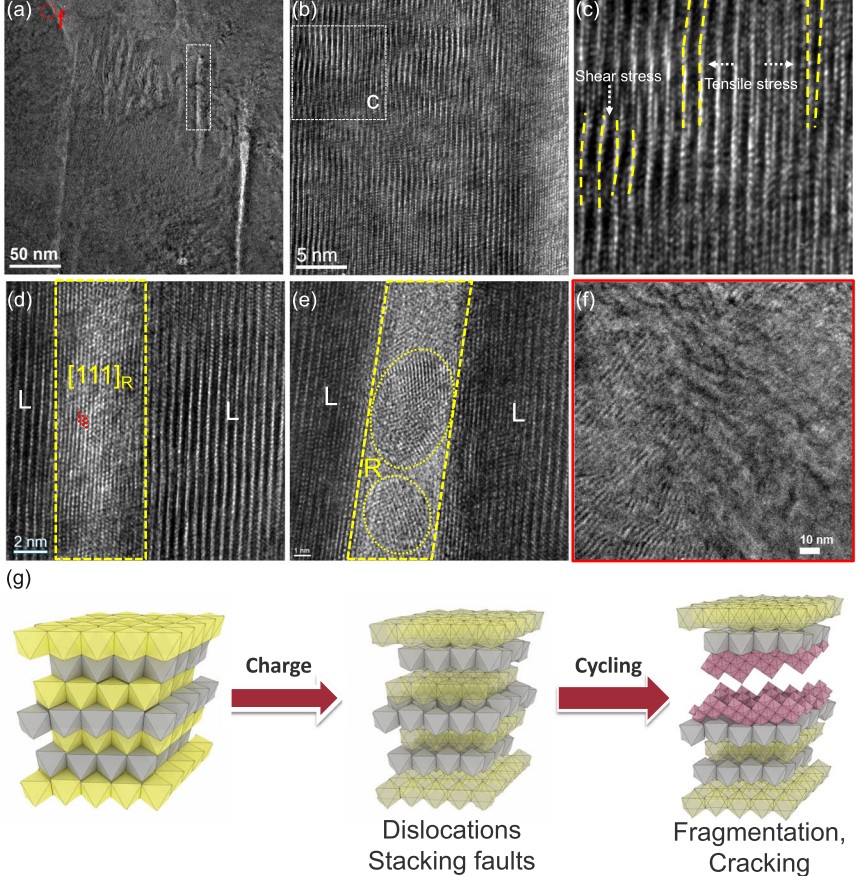

**Fig. 5 TEM characterization on cycled O3 $NaNi_{0.4}Mn_{0.4}Co_{0.2}O_2$ cathode particle (2.0–4.4 V for 100 cycles). a** Low and **b** high magnification TEM image. **c** Zoomed-in view of the dashed square area in **b**. **d**, **e** Zoomed-in views of the dashed rectangle area in **a**. **f** Zoomed-in view of dot-circle area in **a**. **g** Schematic illustration of the structural earthquake during cycling of strained O3 $NaNi_{0.4}Mn_{0.4}Co_{0.2}O_2$ cathode. R and L in **d**, **e** represents rock-salt and layered structure, respectively. Yellow and gray octahedra represents $NaO_6$ and $TMO_6$ of a layered structure, respectively. Purple octahedra means rock-salt structure.

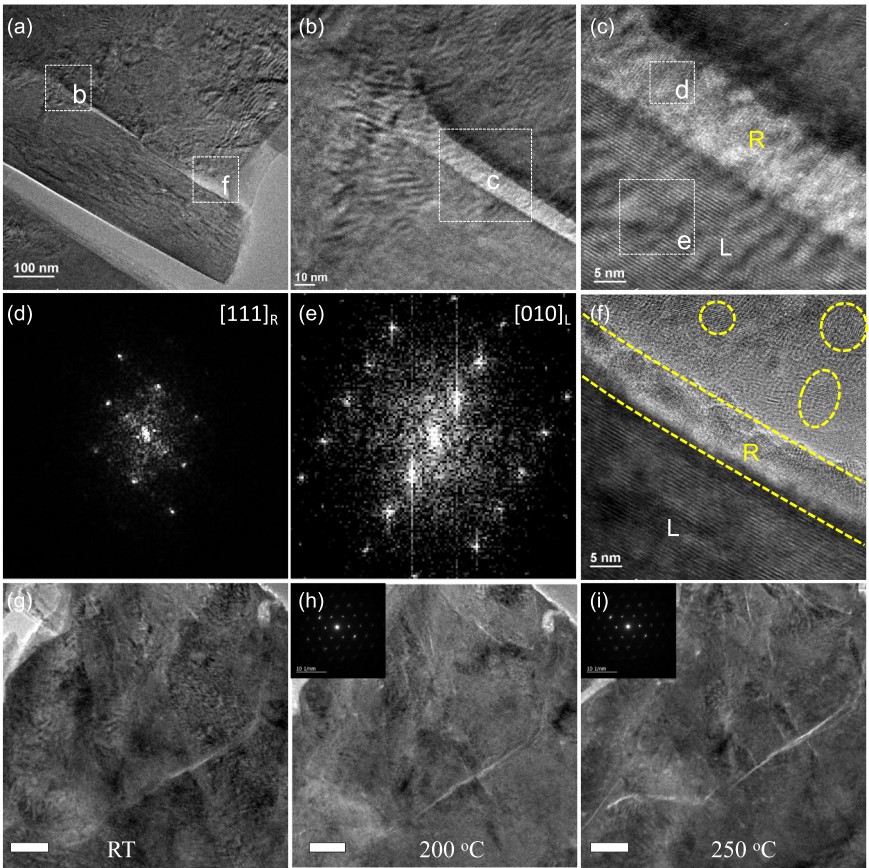

**Fig. 6 TEM characterization on cycled O3 NaNi$_{0.4}$Mn$_{0.4}$Co$_{0.2}$O$_2$ (2.0–3.8 V for 10 cycles). a** Low magnification TEM image. **b** Zoomed-in view of the dashed square area in **a**. **c** Zoomed-in view of the dashed square area in **b**. **d**, **e** Corresponding FFT patterns of the dashed square area in **c**. **f** HRTEM image of the dashed square area **f** in **a**. In situ TEM images of 3.8 V-charged O3 NaNi$_{0.4}$Mn$_{0.4}$Co$_{0.2}$O$_2$ cathode during heating: **g** at room temperature; **h** at 200 °C and **i** at 250 °C. Insets of **h** and **i** are the corresponding SAED patterns. R and L represents the rock-salt and layered structure, respectively.

unrelaxed region (blue circle in Fig. 5a), the curved lattice fringes were preserved, and no formation of cracks or premature cracks can be observed even at the grain boundaries that have long been considered as the preferred crack-initiating sites (Fig. 5f). The structure evolution of the highly strained NaNi$_{0.4}$Mn$_{0.4}$Co$_{0.2}$O$_2$ cathode during charge/discharge is schematically illustrated in Fig. 5g, which is very similar to the earthquake process. The curved lattice fringes of strained layered oxide cathodes introduced by quenching during material synthesis are similar to the curved stratum of the earth due to the stored elastic strain energy. The extraction/insertion of Na$^+$ plays a similar role to plate motion, which leads to the release of stored energy (native strain) in a way of cracking, displacement, and dip/strike faulting that also occurred during the structural degradation of layered cathodes.

Therefore, such strain relaxation does not necessitate a high-voltage charge but should be more related to local strain distribution. Figure 6 shows the structures of cycled strained cathode during charge/discharge with a lower cut-off voltage of 3.8 V for 10 cycles. As shown in Fig. 6a, two strain-relaxed regions can be observed, with one on the surface (marked by dashed square f) and the other one in the interior of the particle (marked by dashed square b). The results are contradictory to the prevailing surface-to-bulk phase transformation mode for layered cathodes, which believed that the degradation was initiated from the surface[61,62]. Although our investigated system is based on a highly strained material, which is different from most of the previously investigated materials, the synthesis of layered oxide cathodes generally involves the sophisticated process, which can

actually induce many different kinds of native structural defects, especially during the large-scale manufacturing process that could easily suffer from temperature heterogeneity. Indeed, boundaries[63–66], nanopores[67], and dislocations[68] have been observed in several lithium-layered oxide cathodes, of which their distribution could act as the starting points of structural degradation, which however does not always initiate from the surface.

The zoomed-in view of square b in Fig. 6a clearly reveals that the strain relaxation was terminated exactly at the end of the strain-unrelaxed region (Fig. 6b). Moreover, Fig. 6c clearly shows that the strain-relaxed region (~10 nm) exhibited rock-salt structure (square d), while the layered structures remained in the strain-unrelaxed region (square e). These are further supported by the corresponding Fast Fourier Transform (FFT) patterns in Fig. 6d, e, which are in good agreement with that projected from [111] direction of rock-salt phase and [010] direction of layered phase, respectively. In another strain-relaxed region (Fig. 6f, zoomed-in view of square f in Fig. 6a), we also observed the formation of rock-salt domains with different orientations, which are similar to those that occurred during high-voltage charge (Fig. 5d, e). Therefore, the charge cut-off voltage does not play a critical role in activating the strain relaxation, which can explain the similar capacity fade behavior of strained NaNi$_{0.4}$Mn$_{0.4}$Co$_{0.2}$O$_2$ cathode during cycling with different cut-off voltage.

To directly stimulate the strain relaxation process and the associated structural transformation during charge/discharge, we further conducted in situ TEM observation during heating of

charged (3.8 V) strained $NaNi_{0.4}Mn_{0.4}Co_{0.2}O_2$ cathode. Before in situ heating, a high distribution of inhomogeneous lattice strain as the contours and intragranular gaps can be clearly observed (Fig. 6g). After heating to high temperature, the pre-existing gaps propagated in accompaniment with the formation of high-density new nanogaps. In particular, we can see that the gaps were widened or appeared starting from the regions that exhibit higher lattice strain (Fig. 6h, i). Nevertheless, the layered structure of the strained $NaNi_{0.4}Mn_{0.4}Co_{0.2}O_2$ cathode was well preserved even after heating to 250 °C, as evidenced by the corresponding SAED patterns in the inset. The results again indicate that layered to rock-salt phase transformation is not the dominant factor to trigger the formation of cracks, but might be a consequence of strain relaxation.

## Discussion

The aforementioned results have clearly emphasized the critical role of native lattice strain in initiating the structural earthquake of sodium-layered oxide cathodes, which can lead to extremely fast capacity degradation. It has clearly distinguished native lattice strain from other common factors such as phase transition and cathode/electrolytes parasitic reactions. Although it is well known that the lattice strain could be also generated due to lattice mismatch and reaction heterogeneity from the aforementioned structural deformation during cycling, it generally takes a long time to break the threshold value and cause an abrupt capacity degradation. Instead, the native lattice strain induced during the synthesis of sodium-layered oxides could significantly exceed the threshold limits, and hence dominate their failure process. The results indicate that it is very important to eliminate the native lattice strain by fine-tuning the synthetic conditions and also to minimize the strain generation during cycling by rational structure tailoring. The concentration and distribution of these lattice strains depend on the quenching conditions (e.g., medium, cooling rate, etc.) Therefore, the selection of quenching medium (e.g., liquid $N_2$, fluid, etc.) and cooling rate/environment that can ensure a homogeneous thermal distribution and transformation could help to alleviate the lattice strain. Moreover, inspired by steel manufacturing, optimization of post-annealing procedures could also help to eliminate these intrinsic lattice strains.

Although the strain relaxation process is not regulated by the voltage window, it is critical to understand the potential factors that can regulate this process. Charge/discharge temperature and rate are two common factors that can dramatically affect the cycle stability of cathode materials. Therefore, we further investigated their influence on the strain relaxation process. Figure 7 compares the structures of cycled strained $NaNi_{0.4}Mn_{0.4}Co_{0.2}O_2$ cathode after 100 cycles of charge/discharge under different conditions. Under a low charge/discharge rate of 0.08C at room temperature, the strain relaxation was relatively smooth, leading to the formation of a straight incision (marked by white rectangles in Fig. 7a, b). This is because the extraction/insertion of $Na^+$ is proceeding in a very slow manner, which can thus minimize the effect of stress heterogeneity. This is similar to common phenomena, such as smooth plate motion or tearing wrinkle paper in a slow manner. Again, we can see the tip of the gap was terminated at the end of the strain-unrelaxed region. With elevated

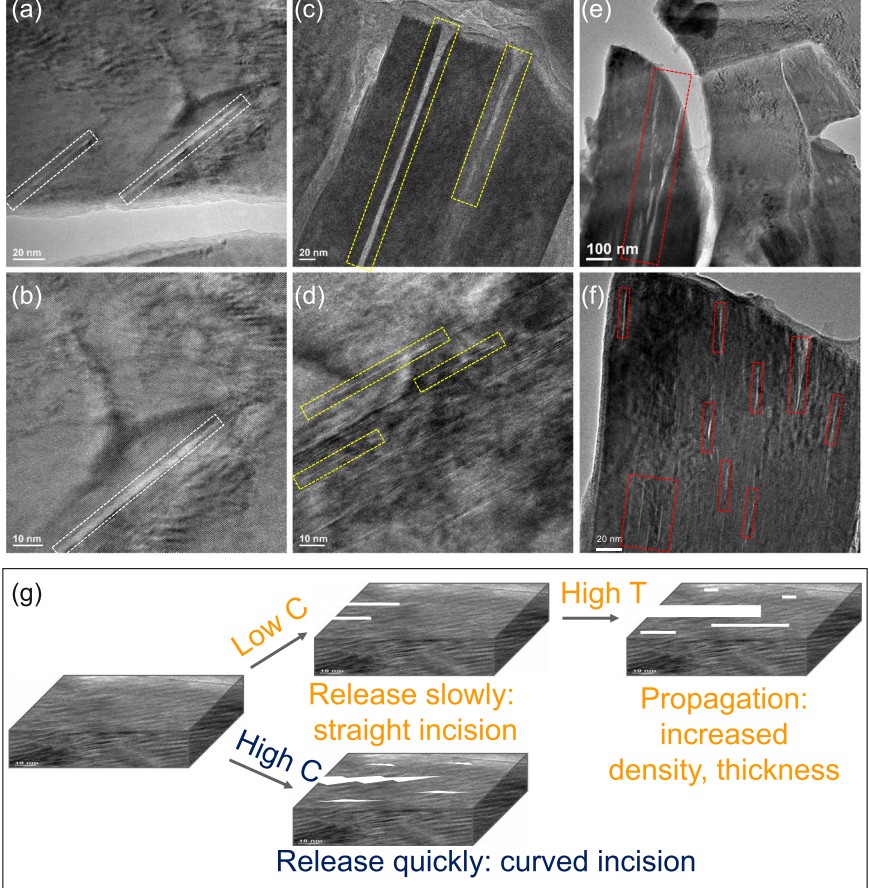

**Fig. 7 Strain relaxation under different conditions.** Low and high magnification TEM images of O3 $NaNi_{0.4}Mn_{0.4}Co_{0.2}O_2$ cathode after 100 cycles within 2.0–4.4 V at different conditions: **a, b** 0.08C under room temperature; **c, d** 0.2C under 55 °C, and **e, f** 1C under room temperature. **g** Schematic illustration on the morphologies of nanogaps in the cycled O3 $NaNi_{0.4}Mn_{0.4}Co_{0.2}O_2$ as a function of cycling condition.

temperature, we can see the propagation and widening of gaps (marked by yellow rectangles in Fig. 7c) as well as the new formation of high-density nanogaps (marked by yellow rectangles in Fig. 7d), which are similar to that observed during in situ TEM heating. This might be because of the increased extraction/insertion kinetic of $Na^+$ at high temperatures. By switching to fast charging/discharging, the strain relaxation is aggressive and vigorous, thus leading to the formation of both large (marked by red rectangles in Fig. 7e) and high-density nanocracks (marked by red rectangles in Fig. 7f) with a curved incision. The relationship between strain relaxation and cycling temperature/rates is schematically illustrated in Fig. 7g. In addition, cracks in the macro scale were also observed in the cycled strained cathode under various conditions due to the relaxation of the native lattice strain (Supplementary Fig. 12).

In summary, through a combination of advanced diagnostics from material synthesis to cell failure, we have discovered the essential role of synthesis-induced native lattice strain in triggering the structure earthquake of sodium-layered oxide cathodes during cycling. The spontaneous relaxation of internal strain that exceeds the threshold limit is the fundamental origin of the abrupt degradation of sodium-layered oxide cathodes. Furthermore, we showed that such a process is not regulated by the charge cut-off voltage, but is strongly coupling with working temperature and charging/discharging rates that can tailor the movement of $Na^+$. Although surface coating or electrolytes modulation is effective in suppressing the irreversible phase transition and mitigating cathode–electrolyte parasitic reactions of sodium-layered oxide cathodes, these approaches cannot prevent the relaxation of intrinsic lattice strain. Our work indicates an urgent need to revisit the crystallographic structure of sodium /lithium layered oxide cathodes such as strain, oxygen vacancies, nanopores, domain boundaries, and other intrinsic defects, and to understand how they affect common battery phenomenon such as capacity fade and voltage decay/hysteresis. Our work also indicates that a rational strain management during cycling of sodium/lithium layered cathodes is required to maximize cycle life.

## Methods

**Material synthesis**. The $Ni_{0.4}Mn_{0.4}Co_{0.2}(OH)_2$ precursor was synthesized through a co-precipitation method. The transition metal sulfate solution (Ni:Mn:Co = 0.4:0.4:0.2 in molar ratio) with a concentration of 2 mol/L was fed into a continuously stirred tank reactor (4 L) under $N_2$ protection. Concentrations of 4 mol/L NaOH and 5 mol/L $NH_3 \cdot H_2O$ were also fed into the tank, respectively. The pH value was controlled at 10.8 during the co-precipitation process by adjusting the NaOH feeding rate[69]. The $Ni_{0.4}Mn_{0.4}Co_{0.2}(OH)_2$ precursor was obtained by filtering and washing with deionized water followed by vacuum drying at 105 °C overnight. To synthesize highly strained O3 $NaNi_{0.4}Mn_{0.4}Co_{0.2}O_2$, $Ni_{0.4}Mn_{0.4}Co_{0.2}(OH)_2$ precursor and NaOH were mixed with a molar ratio of 1:1.05 using a rotary mixer. The mixture powder was then pressed into a pellet and heated to 850 °C for 24 h with a heating rate of 3 °C/min under an air atmosphere. The sample was then quenched from 850 °C to room temperature in air and stored inside a glovebox for further characterization. As a comparison, the P1/P2/O3 $NaNi_{0.4}Mn_{0.4}Co_{0.2}O_2$ was synthesized by naturally cooling from 850 °C to room temperature.

**Electrochemical characterization**. The electrode was made by spreading a mixture of 80 wt.% active material, 10 wt.% C45 and 10 wt.% PVDF (8 wt.% in NMP) onto Aluminum foil. The active material loading was controlled at around 3.5 mg/$cm^2$. The cycling performance of highly strained O3 $NaNi_{0.4}Mn_{0.4}Co_{0.2}O_2$ cathode was evaluated by assembling them into coin cells with Na metal as referenced and a counter electrode inside an Argonne-filled glovebox ($H_2O/O_2$: <0.5 ppm). The electrolyte was 1 mol/L $NaPF_6$ in propylene carbonate with 2 vol% fluoroethylene carbonate additive, and the separator was glass fiber (grade GF/F Glass Microfiber Filter Binder Free, circle, 125 mm). The coin cells were then tested using MACCOR cycler under different voltage window (2.0–4.4 V and 2.0–3.8 V) at different charge/discharge rates and temperatures. These cycled cells were further harvested for ex-situ SEM and TEM analysis. The parasitic reactions of O3 $NaNi_{0.4}Mn_{0.4}Co_{0.2}O_2$ cathode were measured by a home-built high-precision leakage current measurement system[26]. The $Na/NaNi_{0.4}Mn_{0.4}Co_{0.2}O_2$ cell was under formation for two cycles and then charged to different voltages and held at each specific potential for 40 h using a Keithley 2401 source meter. During these processes, the leakage

currents were monitored and measured. The measured leakage current was proportional to the reaction rate of parasitic (side) reactions between the working electrode and the electrolyte.

**In situ SXRD during heating and cycling**. The in situ SXRD experiments were conducted at 11-ID-C of Advanced Photon Sources of Argonne National Laboratory with a wavelength of 0.117418 Å. For the in situ synthesis, the mixture of $Ni_{0.4}Mn_{0.4}Co_{0.2}(OH)_2$ precursor and NaOH was pressed into a 7 mm diameter pellet and loaded into a LINKAM TS-1500 furnace and then heated to 875 °C at a heating rate of 3 °C/min and then held at 875 °C for 7 h followed by quenching to room temperature. For in situ charge/discharge, coin cells with a hole on both top and bottom were used. The holes were sealed with Kapton tape after assembling the $Na/NaNi_{0.4}Mn_{0.4}Co_{0.2}O_2$ coin cells, which were charged/discharged within 2.0–3.8 V and 2.0–4.4 V at 18 mA/g, respectively. SXRD patterns were collected during in situ heating and in situ cycling processes.

**XANES during charge/discharge**. The in situ Ni K-edge XANES experiment was conducted in transmission mode at 20-BM of Advanced Photon Sources of Argonne National Laboratory. The incident beam was monochromatized by using a Si(111) fixed-exit, double-crystal monochromator. During the in situ experiment, the $Na/NaNi_{0.4}Mn_{0.4}Co_{0.2}O_2$ cell was charged/discharged with a constant current density of 25 mA/g between 2.0 and 4.4 V using a MACCOR cycler. Ex-situ Ni K-edge, Co K-edge, and Mn K-edge XANES of strained O3 $NaNi_{0.4}Mn_{0.4}Co_{0.2}O_2$ cathodes under different charge/discharge states were also conducted at 20-BM of Advanced Photon Sources of Argonne National Laboratory. The cycled electrodes were disassembled and rinsed with dimethyl carbonate (DMC, Sigma Aldrich, >99%, anhydrous) to remove electrolyte residue, and then sealed with Kapton tape for measurement.

**Structure characterization**. The TGA analysis was conducted using STA 449 F3 instrument to measure the weight loss of the $Ni_{0.4}Mn_{0.4}Co_{0.2}(OH)_2$/NaOH mixture during heating from room temperature to 900 °C with a heating rate of 10 °C/min under the atmosphere of air. The morphologies of the $NaNi_{0.4}Mn_{0.4}Co_{0.2}O_2$ were characterized by scanning electron microscopy (JEOL 7100F) and TEM (JEOL 2100F). The Zeiss NVision 40 focused ion beam–scanning electron microscopy dual-beam system was used to prepare a cross-section TEM specimen through a standard lift-out procedure. The HAADF imaging and elemental mapping were conducted at FEI Talos F200X (S)TEM equipped with a SuperX energy-dispersive X-ray spectrometer. The In situ heating TEM experiments during heating of a 3.8 V-charged $NaNi_{0.4}Mn_{0.4}Co_{0.2}O_2$ electrode were conducted on JEOL 2100 F from room temperature to 250 °C by using a Gatan bulk heating holder. HREM was carried out on a FEI Titan 80-300 ST, equipped with a CEOS Cc/Cs image corrector and operated at 200 kV. C 1$s$ XPS of slow-cooled layered oxide cathodes were conducted using a PHI 5000 VersaProbe II XPS Microprobe (Physical Electronics), with Al Kα radiation (1486.6 eV, 100 μm diameter at focus, 25 W), $Ar^+$, and electron beam sample neutralization. The obtained spectra were calibrated using the C–C peak at 284.8 eV in the C 1$s$.

**In situ DEMS**. The in situ DEMS experiment was conducted in a custom-made system. The electrode was prepared by spreading a slurry consisting of 80 wt.% $NaNi_{0.4}Mn_{0.4}Co_{0.2}O_2$, 10 wt.% PVDF binder, and 10 wt.% C45 onto a 16-mm diameter carbon paper and then dried (areal active material loading: 10 mg/$cm^2$). We assembled a 2025-coin cell with a hole with a sodium metal anode and glass fiber separator, which was charging/discharging within 2.0–4.4 V at a current density of 30 mA/g. The carrier gas was pre-dehydrated helium (99.999%) with a flow rate of 8 mL/min. The gas species generated during cycling were then passed through a cold trap (mixture of dry ice and ethanol at −78.5 °C and 1 atm) to condense the electrolyte vapor before it was sent to the mass spectrometer (modified 5975C mass-selective detector, Agilent) for analysis.

**Reporting summary**. Further information on research design is available in the Nature Research Reporting Summary linked to this article.

## Data availability

The data that support the findings of this study are available from the corresponding authors G.-L.X., Y.L., and K.A. upon reasonable request.

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

## Acknowledgements

Research at Argonne National Laboratory was funded by the U.S. Department of Energy (DOE) Vehicle Technologies Office. Support from Tien Duong of DOE's Vehicle Technologies Office is gratefully acknowledged. Use of the Advanced Photon Source and Centre for Nanoscale Materials was supported by the U.S. Department of Energy, Office of Science, and Office of Basic Energy Sciences, under Contract No. DEAC02-06CH11357.

## Author contributions

G.-L.X. conceived the idea and designed the experiments. G.-L.X., X.L., Y.R., and Z.C. performed all the in situ SXRD experiments and processed the data. Y.L., X.Z., and G.-L.X. performed the electron microscopy characterization and data analysis. G.-L.X. synthesized the materials and performed the electrochemical characterization. G.-L.X., I.H., and C.-J.S. performed in situ XANES experiments and data analysis. C.Z. and A.D. prepared the samples for ex-situ XAS measurement. Z.Y. conducted the XPS measurement. G.-L.X. wrote and revised the manuscript. G.-L.X. and K.A. managed the project. All authors contributed to discussions and paper revisions.

## Competing interests

The authors declare no competing interests.
