## [Peer Review File · Nature Communications]

REVIEWER COMMENTS

Reviewer #1 (Remarks to the Author):

The manuscript submitted by Amine et al. reports the effect of native lattice strain induced during electrode material synthesis process on the capacity degradation, as revealed by a series of advanced in-situ synchrotron X-ray probes and TEM characterizations. They revealed that the spontaneous relaxation of native lattice strain is responsible for the structural earthquake of layered oxide cathodes during cycling. The data are comprehensive and well present. However, in my personal opinion, the conclusions they draw seem more specific to the investigated system, but not to the com-mon sodium/lithium layered oxide cathodes. This is because the reported electrode $\text{NaNi}_{0.4}\text{Mn}_{0.4}\text{Co}_{0.2}\text{O}_2$ is highly strained, which is different from the widely reported layered electrodes. Therefore, the manuscript may not be suitable to appeal to the readership of high-impact Nature Communications.

Some comments that should be taken into considerations:

1. The authors claimed that the highly reversible Ni oxidation/reduction even at high-voltage charge could not explain the sever capacity loss of strained electrode. However, it is also possible that Co and Mn redox reactions are poorly reversible, leading to the fast capacity fading.
2. It is quite interesting that no O_2 release is observed even charging to 4.4 V, because oxygen redox/release reaction may occur at such high voltage.
3. How does the metal migration during charging contribute to the structural earthquake during cycling?
4. In the introduction part, the authors mentioned that O3 phase is a better cathode candidate than other type of cathodes, which is misleading. This is because both P2 and O3 phases have their own advantages and disadvantages.

Reviewer #2 (Remarks to the Author):

The authors show a convincing argument for the nature of capacity degradation in a sodium layered oxide cathode. However, there are a few inconsistencies that should be clarified to make the paper suitable for publication. The importance of this work couple with the significance of the findings make suitable for publication in Nature Communications. Therefore, I recommend this paper be published in Nature communications pending some minor revisions.

General Comments

A major point that should be addressed is the generality of the statements in the paper for all layered oxide materials. The results presented are only for a sodium material and should not be extended to lithium layered oxides without further experiments showing the same phenomenon in a lithium material.

Some discussion around what causes the high strain in the $O3 \text{NaNi}_{0.4}\text{Mn}_{0.4}\text{Co}_{0.2}\text{O}_2$ structure should be provided. There is a lot of characterization around the fact that the lattice is strained, but no discussion as to why the lattice is strained, how the quenching process induces the strain, or why the lattice is strained. From the TEM images, the strain in the lattice seems to be localized, so any indication on what is different in these regions over the "strain-resistant"? Do new strained regions form during cycling or are they only formed during quenching? Is the strain relaxation process intercalation mediated?

Suggestions on how to alleviate the lattice strain in these materials in the conclusion would also be helpful for further research in this area.

Was any compositional analysis performed on the synthesized material? I am curious if the high strain is resulting from a small sodium deficiency in the structure rather than a true fully occupied $O3$ lattice. Especially since the XRD of the intermediate phase at 350C shows a $P3$ -layered phase.

For the slow cooled phase, you index a few sodium deficient phases, is there also a formation of a sodium excess phase? Or where is the remaining sodium?

Since the slow-cooling process resulted in the formation of several other phases, the O3 material seems to be metastable, does this metastability have any relationship to the high strain seen in its lattice?

For the “strain resisting region”, what are the features of the material in this region that make it strain resistant?

Any comment on why the strain relaxation process is irrespective of voltage window?

Specific Comments

On page 3 line 53, lithium layered oxides can also present in structures where the lithium only resides in tetrahedral sites in ion-exchanged compounds. For clarity, some descriptor for the fact that only octahedral lithium layered oxides presenting in direct synthesis should be added.

On page 6 line 120-122, just because the O3 phase sodium materials begin with the highest Na-content, this fact does not necessarily mean they are the best candidate since other layered structures can have additional sodium electrochemically inserted into the structure if there is a sodium metal anode.

In Figure 1e, it looks like the c lattice parameter decreased during quenching and then increased afterwards. The text in the manuscript on page 8 line 160-162 says otherwise. To this end, are the vacancies/faults/dislocations formed during the quench or during the relaxation after the quench?

The full name of the DEMS measurement is not provided anywhere, just the abbreviation. Please add the full name of the measurements upon the first usage of the abbreviation.

It would be helpful to mention the electrolyte used in the paragraph starting on page 8 line 176 so that the reader does not have to go to the experimental section to get this information.

For the reported layered oxides mention on page 9 line 188-189 (refs 19 and 45), were these materials strained or unstrained?

The XANEs study for nickel oxidation was very nice, but was there any study on the $\text{Co}^{3+}/4+$ redox couple?

Is there a way to quantify the strain in the lattice observed through TEM?

Page 11 line 250, didn't the slow-cooling for the material result in multiple phases? How is the well-aligned lattice fringes for this material relevant compared to the quenched sample if it is multiple phases?

For the inset in Fig 4c, can you be more descriptive on how the inset shows the existence of cation mixing along the c-axis?

The blue circle in Figure 5a is difficult to see, I would suggest changing the color to something more obvious like red or white.

Are there any structural or chemical features that indicate where the cracks/displacement will occur?

At what point does the strain in the lattice relax? Does it start immediately upon deintercalation on first-charge?

Page 14 line 312: Can the results of this material be generalized to all layered oxides that exhibit a surface-to-bulk phase transitions? Or just materials with a highly strained starting lattice?

Page 15 line 352: When you say that the strain generation can be minimized by rational structure tailoring, what types of structural features can be tuned to mitigate the strain? This leads back to my comment about what structural features cause the strain in the first place.

Labeling the regions in Figure 7 discussed in the manuscript on page 16 would be helpful.

During the synthesis, why was the pH maintained at 10.8 during the co-precipitation process? What is significant about this pH value?

All of the instrument details were provided except for the TGA.

A brief descript (1 or 2 sentences) on the high-precision leakage current measurement on page 18 would be helpful.

In Figure 4c, a box around the area where the inset is from would be helpful.

Does the fragmentation/cracking shown in Figure 5g manifest itself on the macro scale? i.e. particle cracking? Or is it only a phenomenon seen in the lattice via TEM?

A point-by-point response to reviewers' comments

Reviewer #1 (Remarks to the Author):

The manuscript submitted by Amine et al. reports the effect of native lattice strain induced during electrode material synthesis process on the capacity degradation, as revealed by a series of advanced in-situ synchrotron X-ray probes and TEM characterizations. They revealed that the spontaneous relaxation of native lattice strain is responsible for the structural earthquake of layered oxide cathodes during cycling. The data are comprehensive and well present. However, in my personal opinion, the conclusions they draw seem more specific to the investigated system, but not to the common sodium/lithium layered oxide cathodes. This is because the reported electrode $\text{NaNi}_{0.4}\text{Mn}_{0.4}\text{Co}_{0.2}\text{O}_2$ is highly strained, which is different from the widely reported layered electrodes. Therefore, the manuscript may not be suitable to appeal to the readership of high-impact Nature Communications.

Response: We thank the reviewer for your comment on our work as “comprehensive and well present”. With all due respect, we respectively do not agree with you about the broad impact of our work. This is because:

- Over the years, most of the failure mechanism understanding on cathode materials has been mostly focusing on structural degradation occurred during operation (e.g. cycling), and the corresponding mitigation strategies have been focused on poste protection such as surface coating. However, synthesis of oxide cathodes generally involves sophisticated process, which can easily induce structural defects in the pristine materials, while the effect of these native structural defects has been overlooked.
- To the best of our knowledge, our work for the first time reported the effect of native lattice strain and the underlying mechanism in sodium layered oxide cathodes, which will for sure stimulate more efforts in understanding and control of other native structural defects in sodium layered oxide cathodes and other battery materials such as lithium transition metal oxide cathodes, solid-state electrolytes and etc. Meanwhile, the native lattice strain induced during quenching is resulted from inhomogeneous temperature field, which might not be frequently observed in lab-scale materials, but could be a common issue during large-scale manufacturing of battery materials that could easily undergo temperature heterogeneity during the calcination process. Indeed, twin boundaries [Ref1] and dislocations[Ref2] has been observed in commercial LiCoO_2 cathodes.
- More importantly, our work also represents an advance to distinguish the contribution of lattice strain, phase transition and parasitic reactions, which have been viewed as the main origins for the degradation of layered oxide cathodes during cycling. However, they are intimately coupling with each other during cycling, making it difficult to understand their explicit role. In this work, by using an artificial strained material as a model material,

we revealed that the lattice strain, either induced during synthesis or accumulated during cycling, plays an overwhelming role in triggering the catastrophic capacity degradation of layered cathodes. Our work highlights the importance of rational strain control in battery materials.

We have added more relevant discussion to clearly describe these points in the revised Introduction. And we believe the findings presented in this work is of great interest to the readership of Nature Communications.

[Ref 1] Jiang, Y. et al., *Atomistic mechanism of cracking degradation at twin boundary of LiCoO₂*. *Nano Energy* 2020, 78, 105364.

[Ref 2] Xu, Z. R. et al., *Charging Reactions Promoted by Geometrically Necessary Dislocations in Battery Materials Revealed by In Situ Single-Particle Synchrotron Measurements*. *Adv. Mater.* 2020, 32, 2003417.

1. The authors claimed that the highly reversible Ni oxidation/reduction even at high-voltage charge could not explain the sever capacity loss of strained electrode. However, it is also possible that Co and Mn redox reactions are poorly reversible, leading to the fast capacity fading.

Response: We thank the reviewer for raising this question. To address your concern, we have conducted *ex-situ* Ni, Co and Mn K-edge X-ray absorption near edge spectroscopy (XANES) on the strained O3 NaNi_{0.4}Mn_{0.4}Co_{0.2}O₂ cathode during charge/discharge process. As shown, all the transition metals (Ni, Co and Mn) undergo reversible oxidation state change even after 10 cycles of charge/discharge that has triggered severe capacity fading in the strained NaNi_{0.4}Mn_{0.4}Co_{0.2}O₂ cathode. Therefore, our results confirmed that the capacity degradation of strained NaNi_{0.4}Mn_{0.4}Co_{0.2}O₂ cathode is not from the cycling-induced structural transformation, but due to the spontaneous relaxation of native lattice strain.

Fig. R1. (a) Ni K-edge, (b) Co K-edge and (c) Mn K-edge XANES of strained O3 NaNi_{0.4}Mn_{0.4}Co_{0.2}O₂ cathode at different charge/discharge state.

2. It is quite interesting that no O₂ release is observed even charging to 4.4 V, because oxygen redox/release reaction may occur at such high voltage.

Response: We thank the reviewer for this comment. Although O₂ gas release has been often observed in the layered oxide cathodes when charged to high-voltage, it is also possible for the transformation of the oxygen species to CO₂ because of the chemical reactions between reactive lattice oxygen and electrolytes at high-voltage. This has been well investigated by Gasteiger group (e.g. J. Phys. Chem. Lett. 2017, 8, 19, 4820-4825; Materials Today 2018, 21, 8, 825-833; 2018 J. Electrochem. Soc. 165 A2869). Indeed, we have observed a significant amount of CO₂ evolution when the voltage goes beyond 3.8 V (Figure 2c).

We have added more discussion and cited these reference in the revised manuscript.

3. How does the metal migration during charging contribute to the structural earthquake during cycling?

Response: Thanks for this comment. The transition metal migration from transition metal layer to sodium or lithium layer has been widely reported in layered oxide cathode system during charge even without the existence of native lattice strain (Science 2014, 343, 519; Angew. Chem., Int. Ed. 2015, 54, 4440; Chem. Mater. 2017, 29, 6684; Energy Environ. Sci. 2015, 8, 2512), which can lead to undesired phase transition and hence capacity/voltage fade. With the existence of native lattice strain, the transition metal cations migration might be exacerbated due to more open atomic structure, which will in turn aggravate the catastrophic earthquake of sodium layered oxide cathodes.

We have added the relevant discussion about the effect of transition metal migration in the revised manuscript.

4. In the introduction part, the authors mentioned that O3 phase is a better cathode candidate than other type of cathodes, which is misleading. This is because both P2 and O3 phases have their own advantages and disadvantages.

Response: Thanks for your comment. We would like to clarify that we were referred to the sodium deficiency of P2-type cathodes, which remains one of the large barriers for their practical application [Small, 2019, 15, 1900233]. We have cited this reference and revised this sentence to avoid misleading as follow:

“Among various phase structures in the sodium layered oxides cathodes, the O3 phase represents the structure with the higher Na content, **which is different from P2-type cathode that has a lower Na content and suffers from sodium deficiency problem.**”

Reviewer #2 (Remarks to the Author):

The authors show a convincing argument for the nature of capacity degradation in a sodium layered oxide cathode. However, there are a few inconsistencies that should be clarified to make the paper suitable for publication. The importance of this work couple with the significance of the findings make suitable for publication in Nature Communications. Therefore, I recommend this paper be published in Nature communications pending some minor revisions.

Response: We would like to thank the reviewer for the recommendation as well as detailed and constructive comments to help us improve the quality of our manuscript. A point-by-point response to your comments is attached below:

1. A major point that should be addressed is the generality of the statements in the paper for all layered oxide materials. The results presented are only for a sodium material and should not be extended to lithium layered oxides without further experiments showing the same phenomenon in a lithium material.

Response: Thanks for the comment. To avoid misleading and comply with the format requirement of Nature Communications, we have revised the title as follows:

“Native Lattice Strain induced Structural Earthquake in **Sodium** Layered Oxide Cathodes”

Meanwhile, we have also revised the relevant text description and discuss the generalization to lithium layered oxide system in a perspective way.

2. Some discussion around what causes the high strain in the $O3 \text{NaNi}_{0.4}\text{Mn}_{0.4}\text{Co}_{0.2}\text{O}_2$ structure should be provided. There is a lot of characterization around the fact that the lattice is strained, but no discussion as to why the lattice is strained, how the quenching process induces the strain, or why the lattice is strained. From the TEM images, the strain in the lattice seems to be localized, so any indication on what is different in these regions over the “strain-resistant”? Do new strained regions form during cycling or are they only formed during quenching? Is the strain relaxation process intercalation mediated?

Response: These are great questions, which certainly should be clearly explained.

Quenching-induced strain generation has been well known in steel manufacturing [*Scripta Materialia* 2013, 68, 321-324]. It has been also often used for the fabrication of battery cathode materials to control grain size, oxygen vacancies and etc.[e.g. *J. Electrochem. Soc.* 2020, 167, 160518; *RSC Adv.*, 2021, 11, 1715-1728] During quenching, a highly inhomogeneous temperature field is generated, which can result in heterogeneous thermal stresses, thus leading

to residual stresses being introduced at the end of the quenching process. Therefore, the concentration and distribution of these lattice strain really depends on the quenching conditions (e.g. medium, cooling rate, etc.) In our experiment, the quenching was conducted under air, which can easily induce temperature heterogeneity in the local lattice region, leading to formation of the observed lattice strain.

Lattice strain could be also accumulated during extensive cycling of layered oxide cathodes, resulting from lattice mismatch between parent phase and new phases. Nevertheless, in the present work, most of the observed lattice strain should be induced during synthesis, rather than during cycling.

We believe the strain relaxation process is (de) intercalation mediated, during which the extraction/insertion of Na^+ play a critical role in triggering the relaxation process of strained O3 $\text{NaNi}_{0.4}\text{Mn}_{0.4}\text{Co}_{0.2}\text{O}_2$ cathodes (please also see response to your comment 8).

We have made them more clearly in the revised manuscript.

3. Suggestions on how to alleviate the lattice strain in these materials in the conclusion would also be helpful for further research in this area.

Response: Thanks for the nice suggestion. As explained above, the concentration and distribution of lattice strain depends on the quenching conditions. Therefore, we believe selection of quenching medium (e.g. liquid N_2 , fluid, etc.) and cooling rate/environment that can ensure a homogeneous thermal distribution and transformation could help to alleviate the lattice strain. Moreover, inspiring from steel manufacturing, optimization of post annealing procedure could also help to eliminate these intrinsic lattice strain.

We have added these suggestions during the discussion.

4. Was any compositional analysis performed on the synthesized material? I am curious if the high strain is resulting from a small sodium deficiency in the structure rather than a true fully occupied O3 lattice. Especially since the XRD of the intermediate phase at 350 °C shows a P3-layered phase.

Response: Thanks for raising this comment. To address your concern, we have conducted high resolution HAADF-STEM element mapping on the pristine strained O3 cathode. As shown, we do not observe a gradient or inhomogeneous Na distribution across the strained region. Although there was a P3 intermediate phase at 350 °C during in situ XRD, pure O3 phase formed when the temperature was further increased to ~ 850 °C (Fig. 1a), which was confirmed by the *ex-situ* XRD of the synthesized final product (Fig. 1g). Therefore, we believe the composition heterogeneity is very limited in the strained region of strained O3 cathode.

Fig. R2. HAADF-STEM element mapping of pristine strained O3 cathode.

5. For the slow cooled phase, you index a few sodium deficient phases, is there also a formation of a sodium excess phase? Or where is the remaining sodium?

Response: Thanks for this question. We want to clarify that the slow cooling experiment in our work was conducted under air atmosphere.

The slow cooling of layered oxide cathode under air will induce formation of Li- or Na-deficient layer and Li_2CO_3 or Na_2CO_3 on the surface of layered oxide cathodes. As reported by Zhang et al. (*Adv. Energy Mater.* 2019, 9, 1901915), CO_2 molecules in the air could be adsorbed onto the exposed Li^+ ions at the solid/gas interface, through two possible adsorption ways: 1) O atoms in CO_2 molecules directly coordinate with Li^+ ions due to the nature of Lewis acid, then external O_2 molecules attack the activated C atoms in CO_2 ; 2) C atoms in CO_2 directly attack the O atoms of LiO_x polyhedral because of the electronegative nature of O atoms. Following these steps, the surficial Li^+ ions will be extracted from layered oxides to form Li_2CO_3 . Driven by the Li^+ concentration gradient from the bulk to the surface, Li^+ ions in the bulk (near the surface) could be continuously extracted, at high rate when the temperature is high, and eventually a Li-deficient layer would be formed in the near-surface region.

We believe a similar process occurred in the slow-cooling sodium layered oxide samples, leading to the formation of Na-deficient phase such as P1 and P2 (Supplementary Fig. 3). To confirm our hypothesis, we have conducted C1s XPS characterization on our slow-cooled sample. As shown, there is a clear peak of O=C-O, which becomes weaker after etching, indicating the existence of Na_2CO_3 on the surface of the slow-cooled sample.

Fig. R3 C1s XPS spectra of slow-cooled sample before and after etching.

6. Since the slow-cooling process resulted in the formation of several other phases, the O3 material seems to be metastable, does this metastability have any relationship to the high strain seen in its lattice?

Response: As explained above, the formation of other Na deficient phase are mostly due to the side reactions between CO_2 and calcinated cathodes, while the formation of native lattice strain are induced by the quenching process resulting from temperature heterogeneity. Therefore, the O3 phase itself is not metastable since it can be obtained by slow cooling in O_2 (*ACS Energy Lett.* 2020, 5, 6, 1718-1725), while the obtained strained O3 structure is metastable due to its high lattice strain.

7. For the “strain resisting region”, what are the features of the material in this region that make it strain resistant?

Response: Thanks for this comment. we would like to clarify that the so-called “strain-resisting region” is referred to the region that has not started the relaxation process upon charge/discharge, rather than the region that is resistant to strain relaxation. To avoid misleading, we have changed it to “strain-unrelaxed region” in the revised manuscript.

8. Any comment on why the strain relaxation process is irrespective of voltage window?

Response: Thanks for this comment.

Practically, as shown in the electrochemical performance (Figure 2b & 2e), we can clearly see that the lower charge cut-off voltage does not help in stabilizing the strained layered cathode. Moreover, we observed similar strain relaxation process during cycling of strained cathode within both 2.0-4.4 V (Figure 5) and 2.0-3.8 V (Figure 6).

Fundamentally, during charge/discharge of sodium layered oxide cathodes, Na⁺ generally diffuse along either “octahedral-tetrahedral-octahedral” (O phase) or “prismatic-prismatic” (P phase). With the existence of native lattice strain, the TMO₆ and NaO₆ octahedra are both highly strained. Therefore, the extraction/insertion of Na⁺ will need to overcome extra energy barrier, which might be responsible for the strain relaxation. Meanwhile, the strained TM and Na layer might decrease oxygen release energy, leading to the evolution of O₂ and formation of cracks and further facilitate the strain relaxation process.

Nevertheless, higher cut-off voltage might trigger more O₂ formation and severe parasitic reactions, which could accelerate the strain relaxation process. Therefore, to be more accurate, we have revised our description to “**strain relaxation process is not regulated by the voltage window**” in the revised manuscript.

9. On page 3 line 53, lithium layered oxides can also present in structures where the lithium only resides in tetrahedral sites in ion-exchanged compounds. For clarity, some descriptor for the fact that only octahedral lithium layered oxides presenting in direct synthesis should be added.

Response: Thanks for pointing out this inconsistency. We agree with the reviewer that some lithium layered oxide cathode could have Li⁺ and transition metal cations in the tetrahedral sites, such as LiFeO₂ through ion-exchange (*J. Am. Chem. Soc.* 2008, 130, 3554-3559). Therefore, we have revised the description as follows:

“Compared with their Li analogs that present **mostly** the octahedral structure **through direct synthesis**, sodium layered oxide cathodes can be classified into P-type (prismatic) and O-type (octahedral), depending on the surrounding Na environment and the number of unique oxide layers.”

10. On page 6 line 120-122, just because the O3 phase sodium materials begin with the highest Na-content, this fact does not necessarily mean they are the best candidate since other layered structures can have additional sodium electrochemically inserted into the structure if there is a sodium metal anode.

Response: We apologize for the inaccurate statement. We would like to clarify that we were referred to the sodium deficiency of P2-type cathodes, which remains one of the large barriers

for their practical application [Small, 2019, 15, 1900233]. To avoid misleading, we have cited this reference and revised the sentence as follows:

“Among various phase structures in the sodium layered oxides cathodes, the O3 phase represents the structure with the higher Na content, **which is different from P2-type cathode that has a lower Na content and suffers from sodium deficiency problem.**”

11. In Figure 1e, it looks like the c lattice parameter decreased during quenching and then increased afterwards. The text in the manuscript on page 8 line 160-162 says otherwise. To this end, are the vacancies/faults/dislocations formed during the quench or during the relaxation after the quench?

Response: Thanks for this comment to allow us to describe the process more clearly. The a and c lattice parameter both decreased when the temperature of the *in situ* furnace was abruptly decreased from 875 °C to room temperature (about 30 minutes), which are both due to the lattice shrinkage when exposed to low temperature. Afterwards, the material undergo a similar process as desodiation of layered oxide cathodes during charge, during which the a lattice parameter was continuously decreased while c lattice parameter was increased dramatically during the relaxation process. This is because although the heated powder material was exposed to a low temperature environment abruptly, the inhomogeneous thermal stress still need some time to be completely released. Therefore, we believe the formation of vacancies/stacking faults/dislocations occurred during the relaxation process.

We have revised the text in the revised manuscript.

12. The full name of the DEMS measurement is not provided anywhere, just the abbreviation. Please add the full name of the measurements upon the first usage of the abbreviation.

Response: Thanks for this suggestion. We have revised DEMS as “differential electrochemical mass spectrometry (DEMS)” when it appeared for the first time.

We have also checked other acronym in the revised manuscript.

13. It would be helpful to mention the electrolyte used in the paragraph starting on page 8 line 176 so that the reader does not have to go to the experimental section to get this information.

Response: Thanks for the suggestion. We have added the formulation of electrolytes when we first discussed the electrochemical performance.

“The electrochemical performance of the as-prepared cathode was evaluated by using half cells with sodium metal as referenced and counter electrode. The electrolyte was 1 mol/L NaPF₆ in propylene carbonate with 2 vol% fluoroethylene carbonate additive.”

14. For the reported layered oxides mention on page 9 line 188-189 (refs 19 and 45), were these materials strained or unstrained?

Response: Thanks for this question. Ref. 19 was related to a commercial lithium NCM333 cathode, which however does not provide their synthesis process and TEM image. Thus, we cannot determine if it is strained not. To avoid uncertainty, we have removed this reference.

For Ref. 45, the material was synthesis through high-temperature calcination followed by slow cooling, which should be unstrained.

15. The XANEs study for nickel oxidation was very nice, but was there any study on the Co^{3+/4+} redox couple?

Response: Thanks for the comment. To address your concern, we have conducted ex-situ Ni, Co and Mn K-edge on the strained O3 NaNi_{0.4}Mn_{0.4}Co_{0.2}O₂ cathode during charge/discharge process. As shown, all the transition metals (Ni, Co and Mn) undergo reversible oxidation state change even after 10 cycles of charge/discharge that has triggered severe capacity fading in the strained O3 NaNi_{0.4}Mn_{0.4}Co_{0.2}O₂ cathode. Therefore, our results confirmed that the capacity degradation of strained NaNi_{0.4}Mn_{0.4}Co_{0.2}O₂ cathode is not from the cycling-induced structural transformation, but due to the spontaneous relaxation of native lattice strain.

Figure R1. (a) Co K-edge and (b) Mn K-edge of strained O3 NaNi_{0.4}Mn_{0.4}Co_{0.2}O₂ cathode at different charge/discharge state.

16. Is there a way to quantify the strain in the lattice observed through TEM?

Response: Thanks for this constructive suggestion. The quantification of lattice strain is possible by conducting 4D-STEM on a high-end instrument. Unfortunately, we lack the access to this type of capabilities owing to the pandemic. On the other hand, the combination of synchrotron XRD and our quantitative TEM characterizations and the correlation to the materials' performance is good enough to support our conclusion of the strain impact on the structure of this material. We hope our response can resolve your concern.

17. Page 11 line 250, didn't the slow-cooling for the material result in multiple phases? How is the well-aligned lattice fringes for this material relevant compared to the quenched sample if it is multiple phases?

Response: We would like to clarify that a critical difference between our slow-cooling (under air) and the ref. 48 (under O₂) is the atmosphere.

The slow cooling of layered oxide cathode under air will induce formation of Li- or Na-deficient layer and Li₂CO₃ or Na₂CO₃ on the surface of layered oxide cathodes. As reported by Zhang et al. (*Adv. Energy Mater.* 2019, 9, 1901915), CO₂ molecules in the air could be adsorbed onto the exposed Li⁺ ions at the solid/gas interface, through two possible adsorption ways: 1) O atoms in CO₂ molecules directly coordinate with Li⁺ ions due to the nature of Lewis acid, then external O₂ molecules attack the activated C atoms in CO₂; 2) C atoms in CO₂ directly attack the O atoms of LiO_x polyhedral because of the electronegative nature of O atoms. Following these steps, the surficial Li⁺ ions will be extracted from layered oxides to form Li₂CO₃. Driven by the Li⁺ concentration gradient from the bulk to the surface, Li⁺ ions in the bulk (near the surface) could be continuously extracted, at high rate when the temperature is high, and eventually a Li-deficient layer would be formed in the near-surface region. We expect a similar process occurred in the slow-cooling sodium layered oxide samples, leading to formation of Na-deficient phase such as P1 and P2 (Supplementary Fig. 3). To confirm our hypothesis, we have conducted C1s XPS characterization on our slow-cooled sample. As shown, there is a clear peak of O=C-O, which becomes weaker after etching, indicating the existence of Na₂CO₃ on the surface of the slow-cooled sample.

Fig. R3 C1s XPS spectra of slow-cooled sample before and after etching.

18. For the inset in Fig 4c, can you be more descriptive on how the inset shows the existence of cation mixing along the c-axis?

Response: Thanks for this suggestion. In the inset of Fig. 4c, we can see the overlap of Na layer and TM layer, which may be a sign of cation mixing. Nevertheless, because the lattice is tilted, it is hard to clearly distinguish cation mixing and lattice distortion. Therefore, to be accurate, we have revised the description to “**lattice distortion**”.

19. The blue circle in Figure 5a is difficult to see, I would suggest changing the color to something more obvious like red or white.

Response: Thanks for the suggestion. We have changed the color to red and added a label of “F” with the circle.

20. Are there any structural or chemical features that indicate where the cracks/displacement will occur?

Response: Thanks for this question. As shown in the in situ TEM heating experiment (Fig. 6g-6i), the cracks/displacement were exacerbated at the region with higher native lattice strain. Therefore, we believe that the cracks/displacement would prefer to initiate from the high lattice strained region.

21. At what point does the strain in the lattice relax? Does it start immediately upon deintercalation on first-charge?

Response: Thanks for this comment. The quenched O3 material with high native lattice strain is metastable, which will initiate the strain relaxation upon the extraction of Na⁺. Please see our response to your comment 2, 6 and 8.

22. Page 14 line 312: Can the results of this material be generalized to all layered oxides that exhibit a surface-to-bulk phase transitions? Or just materials with a highly strained starting lattice?

Response: This a good question. In our opinion, the conventional surface-to-bulk degradation mode is based on defect-free crystals. However, the synthesis of layered oxide cathodes involve sophisticated process. Indeed, increasing studies have confirmed the existence of native structural defects in the pristine battery materials, which could affect their electrochemical behavior (Ref 1-6). The distribution of these structural defects within the battery material particles will determine the initial location of structural degradation, which however does not always initiate from the surface.

Therefore, we believe that the degradation mode reported in this work could be generalized to most of layered oxide cathodes since they could contain various native structural defects such as boundaries, nanopores, dislocations and etc.

[Ref1] Lee, S. Y. et al., *Revisiting Primary Particles in Layered Lithium Transition-Metal Oxides and Their Impact on Structural Degradation*. *Adv. Sci.* 2019, 6, 1800843.

[Ref 2] Jiang, Y. et al., *Atomistic mechanism of cracking degradation at twin boundary of LiCoO₂*. *Nano Energy* 2020, 78, 105364.

[Ref 3] Li S. et al., *Direct Observation of Defect-Aided Structural Evolution in a Nickel-Rich Layered Cathode*. *Angew. Chem. Int. Ed.* 2020, 59, 22092-22099.

[Ref 4] Ahmed, S. et al., *Understanding the formation of antiphase boundaries in layered oxide cathode materials and their evolution upon electrochemical cycling*. *Matter* 2021, <https://doi.org/10.1016/j.matt.2021.10.001>.

[Ref 5] Ahmed, S. et al., *The Role of Intragranular Nanopores in Capacity Fade of Nickel-Rich Layered Li(Ni_{1-x-y}Co_xMn_y)O₂ Cathode Materials*. *Acs Nano* 2019, 13, 10694-10794.

[Ref 6] Xu, Z. R. et al., *Charging Reactions Promoted by Geometrically Necessary Dislocations in Battery Materials Revealed by In Situ Single-Particle Synchrotron Measurements*. *Adv. Mater.* 2020, 32, 2003417.

23. Page 15 line 352: When you say that the strain generation can be minimized by rational structure tailoring, what types of structural features can be tuned to mitigate the strain? This leads back to my comment about what structural features cause the strain in the first place.

Response: Thanks for this comment. Please see our response to your comment 3.

24. Labeling the regions in Figure 7 discussed in the manuscript on page 16 would be helpful.

Response: Thanks for the suggestion. We have marked the regions in the revised Figure 7 for the referred text description.

25. During the synthesis, why was the pH maintained at 10.8 during the co-precipitation process? What is significant about this pH value?

Response: The pH value during coprecipitation process of transition metal hydroxides could significantly affect the formation of primary and secondary particles (e.g. size of secondary particles). It is generally controlled at ~ 9-12 with the purpose to simultaneously ensure the presence of sufficient amount of hydroxyl anions within the solution and ensure good solubility of the metal ammonia complex, both required for successful (pH>7) and uniform (pH<12) precipitation of transition metal hydroxides (*J. Phys. Chem. B* 2019, 123, 3291-3303).

In this work, the pH value was controlled at 10.8 to control the secondary particle size at 10-12 μm , which for sure could be changed but is however out of the scope of this work.

We have cited these reference in the Methods.

26. All of the instrument details were provided except for the TGA.

Response: Thanks for pointing it out. We have added the instrument detail (STA 449 F3 instrument) of TGA in the Methods.

27. A brief descript (1 or 2 sentences) on the high-precision leakage current measurement on page 18 would be helpful.

Response: Thanks for the suggestion. We have added the relevant discussion in the revised manuscript as follows:

“The parasitic reactions of O3 $\text{NaNi}_{0.4}\text{Mn}_{0.4}\text{Co}_{0.2}\text{O}_2$ cathode were measured by a home-built high-precision leakage current measurement system.²⁶ The $\text{Na}/\text{NaNi}_{0.4}\text{Mn}_{0.4}\text{Co}_{0.2}\text{O}_2$ cell was under formation for two cycles and then charged to different voltage and held at each specific potential for 40 h using a Keithley 2401 source meter. During these process, the leakage currents were monitored and measured. The measured leakage current was proportional to the reaction rate of parasitic (side) reactions between the working electrode and the electrolyte.”

28. In Figure 4c, a box around the area where the inset is from would be helpful.

Response: Thanks for the suggestion. We have inserted a box to mark the enlarged area for the inset.

29. Does the fragmentation/cracking shown in Figure 5g manifest itself on the macro scale? i.e. particle cracking? Or is it only a phenomenon seen in the lattice via TEM?

Response: Thanks for this comment. The fragmentation and cracking does exist in multiscale. We have provided the FIB cross-sectional SEM image of cycled O3 $\text{NaNi}_{0.4}\text{Mn}_{0.4}\text{Co}_{0.2}\text{O}_2$ cathode at various condition, which clearly showed the existence of intergranular cracking within the secondary particles.

Fig. R4. FIB cross-sectional SEM images of O3 $\text{NaNi}_{0.4}\text{Mn}_{0.4}\text{Co}_{0.2}\text{O}_2$ cathode at various charge/discharge conditions: (a) pristine; (b) 0.08C for 100 cycles at RT; (c) 0.2C for 100 cycles at 55 °C; (d) 1C for 100 cycles at RT.

REVIEWERS' COMMENTS

Reviewer #2 (Remarks to the Author):

The authors have adequately addressed all of the reviewers comments. This manuscript is sufficient for publication and will make an excellent contribution to Nature Communications.